# Application of Various Machine Learning Models for Process Stability of Bio-Electrochemical Anaerobic Digestion

Ain Cheon [1], Jwakyung Sung [2], Hangbae Jun [1], Heewon Jang [3], Minji Kim [1] and Jungyu Park [3],*

[1] Department of Environmental Engineering, Chungbuk National University, Cheongju 28644, Korea; ain722@naver.com (A.C.); jhbcbe@cbnu.ac.kr (H.J.); qhfhfhwkd12@naver.com (M.K.)
[2] Department of Crop Science, Chungbuk National University, Cheongju 28644, Korea; jksung73@chungbuk.ac.kr
[3] Department of Advanced Energy Engineering, Chosun University, Gwangju 61457, Korea; hwjang@chosun.ac.kr
* Correspondence: jp@chosun.ac.kr; Tel.: +82-62-230-7119; Fax: +82-62-210-7111

**Abstract:** The application of a machine learning (ML) model to bio-electrochemical anaerobic digestion (BEAD) is a future-oriented approach for improving process stability by predicting performances that have nonlinear relationships with various operational parameters. Five ML models, which included tree-, regression-, and neural network-based algorithms, were applied to predict the methane yield in BEAD reactor. The results showed that various 1-step ahead ML models, which utilized prior data of BEAD performances, could enhance prediction accuracy. In addition, 1-step ahead with retraining algorithm could improve prediction accuracy by 37.3% compared with the conventional multi-step ahead algorithm. The improvement was particularly noteworthy in tree- and regression-based ML models. Moreover, 1-step ahead with retraining algorithm showed high potential of achieving efficient prediction using pH as a single input data, which is plausibly an easier monitoring parameter compared with the other parameters required in bioprocess models.

**Keywords:** machine learning; bio-electrochemical anaerobic digestion; methane yield; pH; process stability





## 1. Introduction

Anaerobic digestion (AD) is gaining attention as a promising technology for biogas production from various organic wastes, such as food waste, waste activated sludge, livestock manure, and landfill leachate [1]. However, AD performances are often affected by substrate characteristics, organic loading rate (OLR), accumulated volatile fatty acids (VFAs) concentration, pH, alkalinity, ammonia concentration, and toxic substances [2,3]. Therefore, AD reactors occasionally exhibit unstable methane production and inefficient organic degradation rate [4,5]. In particular, highly concentrated and easily biodegradable organic matter, such as food waste, interrupts efficient methane production and fast stabilization by accelerating VFA accumulation and pH decrease, resulting in an imbalance between acidogenesis and methanogenesis [6].

Bio-electrochemical anaerobic digestion (BEAD) is gaining attention as an advanced technology that improves microbial activity and growth rates as well as organic removal efficiency and biogas productivity by supplying low voltage (0.2~1.0 V) through bio-electrodes in an AD reactor [7,8]. BEAD systems are superior to AD systems with respect to organic substances removal and biogas production, and that a decrease in pH and VFA accumulation has a low inhibitory effect on methane production [9–11]. Previous lab-scale studies have sufficiently demonstrated the superiority of BEAD through basic studies such as reaction mechanism identification, changes in microbial community structure, electrode configuration, and material suitability [12–14].

Operational stability should be examined as the next step to enhance the applicability of BEAD because operational stability and maintainability of BEAD are important parameters for its application in full-scale processes. This can be achieved by predicting the performance based on the long-term performance of BEAD. In BEAD processes, the analytical parameters are nonlinear in nature [15]. Various methods for forecasting process performance have been researched to improve operational stability by analyzing nonlinear patterns.

Machine learning (ML), a statistical forecasting method, is gaining significant attention for forecasting performance and preventing operational risks. ML can be successfully applied into process models because of its capability to interpret the nonlinear relationships that might be produced among variables (multi input/output) in a complex system [16]. Compared with the AD models, ML can model and predict complex relationships between dependent and independent variables associated with the AD process, without requiring detailed mechanisms of anaerobic processes [17]. In addition, ML models contain a class of generic nonlinear regression models that learn the arbitrary mapping of the input data to the output data to obtain computational models with high predictive accuracy [18]. Hence, an extensive understanding in process model is not required in ML modeling [19]. This suggests that ML can support the long-term process stability of BEAD by applying some operational parameters as input data. BEAD is proven techniques that could achieve a higher process stability than that of conventional AD, supporting bio-electrochemically active microorganism and preventing various inhibitions that cause failure of a reaching steady state [8,20]. Based on these advantages of BEAD, various ML models could be applied into long-term operation of BEAD process for supporting operational stability and accelerating biogas production. However, in-depth study results supporting the long-term process stability of BEAD have not yet been reported, highlighting the need for studying ML applications of BEAD.

Conventional ML models focused on raw data collected during specific operational periods for prediction of future performances by using simultaneous prediction method [21]. Although that method was widely applied to continuously operated bio-process, simultaneous prediction has a limitation in applying new input data that is continuously accumulated. A 1-step ahead algorithm showed a possibility of continual training which contributes an achieving a higher adoption to bio-process. Previous study clearly showed that the 1-step ahead with retraining algorithm was suitable for the practical application by predicting performances derived from continuously operated bio-process [21].

Therefore, a practical application of ML to BEAD for treating food waste was suggested in this study using a long-term evaluation of the effects of operational parameters on BEAD reactors. Various ML models with multi-step and 1-step ahead algorithms were applied to forecast the performance and achieve high operational stability of the BEAD. Moreover, pH was applied as a single input data to evaluate the possibility of real-time prediction and practical applications. The 1-step ahead method, which utilized prior data of BEAD performances, could enhance the prediction accuracy. In addition, 1-step ahead with the retraining algorithm could achieve high prediction accuracy when pH was used as a single input parameter.

## 2. Materials and Methods

### 2.1. Data Preprocessing

The data used in this study were collected from a lab-scaled single-chamber BEAD reactor (effective volume: 20 L) treating food waste. The BEAD reactor was operated for 1086 days under various organic loading rates (OLRs) based on the input chemical oxygen demand (COD) concentration. The details of the BEAD reactor have been published in previous studies [22,23]. The pH, alkalinity, and COD removal efficiency were used as the input parameters, and the input COD based methane yield (L-$CH_4$/g-COD) was used as the output parameter. The input parameters were chosen in accordance with the variable importance analysis results. When pH, alkalinity, and COD removal efficiency were applied as independent variable, the highest $R^2$ value was calculated. The lab-scaled

BEAD reactor was operated by supplying voltage of 0.3 V under gradually increased OLRs (Table 1). During stage 1, the BEAD reached intermediate and final steady states after 98 and 250 days, respectively, of operation and continued stable methane production by stage 5. Stable methane yields in the BEAD reactor at the final steady state of S1–S5 (2.0–10.0 kg/m$^3$·d) were $0.35 \pm 0.02$, $0.36 \pm 0.04$, $0.36 \pm 0.04$, $0.36 \pm 0.02$, and $0.36 \pm 0.02$ L-CH$_4$/g-COD, respectively. More details on BEAD performance are presented in Table 1.

**Table 1.** Methane production and yield in BEAD reactor during the total operation periods.

| Item | Stage 1 | Stage 2 | Stage 3 | Stage 4 | Stage 5 |
|---|---|---|---|---|---|
| Operation period (days) | 0–365 | 366–598 | 599–795 | 796–950 | 951–1086 |
| OLR (kg-COD/m$^3$·d) | $2.5 \pm 0.6$ | $1.0 \pm 0.2$ | $6.0 \pm 0.3$ | $8.0 \pm 0.3$ | $10.0 \pm 0.4$ |
| pH | $7.7 \pm 0.3$ | $8.0 \pm 0.2$ | $8.1 \pm 0.1$ | $8.1 \pm 0.1$ | $8.2 \pm 0.1$ |
| Alkalinity (g/L as CaCO$_3$) | $7.6 \pm 0.9$ | $10.1 \pm 0.8$ | $13.9 \pm 0.8$ | $14.8 \pm 0.7$ | $15.3 \pm 0.7$ |
| Total VFAs (mg/L) | $2.6 \pm 0.9$ | $3.1 \pm 0.2$ | $3.9 \pm 0.2$ | $4.6 \pm 0.3$ | $5.3 \pm 0.3$ |
| COD removal efficiency (%) | $67.8 \pm 7.2$ | $71.4 \pm 2.5$ | $73.5 \pm 3.0$ | $75.1 \pm 2.3$ | $76.3 \pm 1.7$ |
| CH$_4$ production (L/day) | $15.7 \pm 4.6$ | $33.9 \pm 3.9$ | $51.2 \pm 6.3$ | $63.4 \pm 3.9$ | $74.7 \pm 3.4$ |
| CH$_4$ yield (L-CH$_4$/g-COD) | $0.32 \pm 0.07$ | $0.35 \pm 0.04$ | $0.35 \pm 0.04$ | $0.36 \pm 0.02$ | $0.36 \pm 0.01$ |

BEAD: bio-electrochemical anaerobic digestion, OLR: organic loading rate, VFA: volatile fatty acid, COD: chemical oxygen demand.

*2.2. Statistical Analysis*

2.2.1. Principal Component Analysis (PCA)

The PCA analysis was conducted using pH, alkalinity, COD removal efficiency, and methane yield of the BEAD reactor as principal components. The axes of principal components presenting eigenvalues of 1.0 that showed the dispersion size of orthogonal data were considered when the number of principal components was determined [24]. The varimax rotation method that can explain the relationships between variables and components was used to rotate the axis [21]. The Bartlett's sphericity test and the Kaiser Meyer Olkin (KMO) test were applied to determine validity of preprocessed data for the PCA. The KMO test results reveal the degree of covariance between the variables used in the analysis and the components inherent in the data. As the degree of covariance approaches 1, the validity of the analysis is high, and the analysis can be performed only when it is at least 0.5 [25]. Statistical analysis was performed using four variables that satisfied the standard value of KMO. The KMO-value and *p*-value of four variables which consist of pH, alkalinity, COD removal efficiency, and methane yield were 0.73 and less than 0.01, respectively.

2.2.2. Variable Importance Analysis

Input data that was properly selected simplifies the model algorithm and improves its applicability to full scale processes. Therefore, Recursive feature elimination (RFE) was used to remove low important variables, one at a time. The lowest RMSE of 0.2382 L-CH$_4$/g-COD and the highest R$^2$ of 0.971 were obtained when the three independent variables (ranked as follows: pH > COD removal efficiency > alkalinity) were applied. Therefore, these three parameters were used as input data in ML models used in this study.

*2.3. ML*

2.3.1. Prediction Models

The input layer treats all the input data by communicating with the external environment that provides significant pattern [26]. These input data are transferred to the hidden layer, and every input neuron could show independent variables that can affect to the outputs of the neural network (Figure 1a). The hidden layer collects those neurons that include applied activation function. Because hidden layer processes the inputs obtained from previous layer, it is responsible for extracting the required features from the input data [27]. The output layer collects and transmits information according to a designated method.

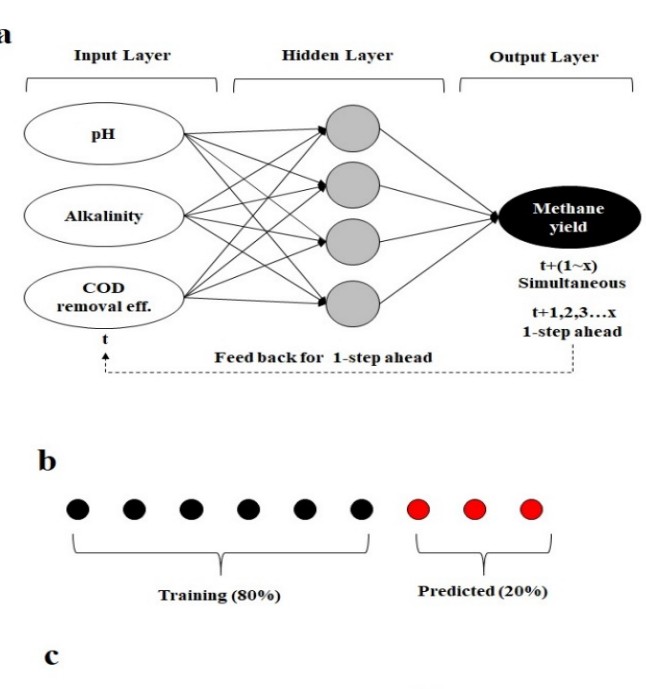

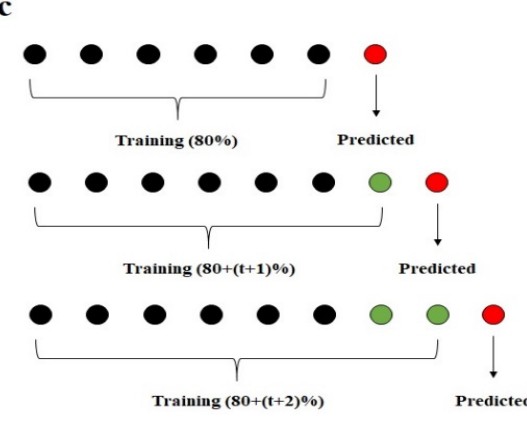

**Figure 1.** Schematic diagrams for understanding the (**a**) machine learning algorithm, (**b**) multi-step ahead method, and (**c**) 1-step ahead with retraining method [21].

The following five ML models were applied to predict the methane yield of BEAD reactor: random forest (RF), extreme gradient boosting (XGboost), support vector regression (SVR), long short-term memory (LSTM), and recurrent neural network (RNN). A neural network algorithm has three different layers: input, hidden, and output [26,27].

Each ML was modeled by using multi-step ahead method and 1-step ahead with retraining method. More detailed fundamentals of each method are presented in Section 2.3.3 and Section 2.3.4, respectively. This study used R program (version 3.5.1), which is a software for statistical analysis, ML modeling, and graphics formation. The R program packages used for each ML model are listed in Table 2.

**Table 2.** R program packages used for the prediction of methane yield.

| ML Models | Packages |
| --- | --- |
| Random Forest (RF) | Package "randomForest" |
| Extreme gradient boosting (XGboost) | Package "rxgboost" |
| Support Vector Regression (SVR) | Package "e1071" |
| Long Short-Term Memory (LSTM) | Package "rnn" and "keras" |
| Recurrent Neural Networks (RNN) | Package "rnn" |

2.3.2. Validations and Model Accuracy Calculation

Determining the optimal model parameters is important for improving the prediction accuracy of ML models [27]. Cross validation was introduced to determine the optimal combinations of hyperparameters. Learning rate, number of hidden nodes, batch size for LSTM and RNN, C and sigma for SVR, and number of trees for the RF and XGboost were considered as hyperparameters to optimize each model [21]. The 10-fold cross–validation was repeated three times to prevent overfitting and evaluate the prediction performance. The data was divided into a training set and a test set, which were used for the model construction and evaluation of prediction accuracy.

Based on the continuously accumulated operation data of the BEAD reactor during operational stages 1–5, 80% of the total time-series data were provided as training data, and the posterior 20% was provided as test data (Figure 1b,c). For predicting final methane yield, pH, alkalinity, and COD removal efficiency were used as input parameters and amount of training and predicting samples were 312ea (80% of operation period) and 78ea (20% of operation period), respectively. To compare the prediction accuracies of each ML model, the RMSEs of all ML model results were evaluated in this study, using Equation (1):

$$\text{RMSE} = \sqrt{\frac{1}{n}\sum_{i=1}^{n}(y_i - \hat{y}_i)^2} \tag{1}$$

2.3.3. Multi-Step Ahead Method

The raw data (see Supplementary Materials) obtained by BEAD reactor operation for 3 years was divided into training and test datasets (Figure 1). In this study, 80% of the raw data were used for training, and the remaining 20% of raw data were used for testing. The multi-step ahead method was applied using split-sample experiments [21]. After modeling was finished, the prediction accuracy was evaluated by comparing with predicted values and known data. Therefore, multi-step ahead prediction was performed by using only 80% past data of raw dataset as inputs for training process.

2.3.4. 1-Step Ahead with the Retraining Method

In contrast with multi-step ahead method, 1-step ahead with the retraining method considers the previous learning contents required in the time-series data analysis and updates the inputs sequentially for the retraining process. In the 1-step ahead with the retraining method, the network trained up to past time step n th is retrained to predict the outputs for the next time step, that is, the (n + 1) th step (Figure 1c) [28,29]. Cumulative 1-step ahead retraining and learning were performed as follows: a model using the data at time point t was constructed and the future value at time t + 1 was predicted. After adding the data at time t + 1, a new model was built to retrain data at [1, ... , t + 1] to predict the value at time step t + 2. After repeating this process and when predicting the value after time N elapsed, the model is constructed using data from the time step [1, ... , t + 1, ... , t + n], and the value is predicted at time step t + n + 1 (Figure 2b) [27]. In this study, when time t − 3 was included, the prediction accuracy was the highest. Therefore, input parameters and predicted outputs at t − 3 step was applied for retraining process of each 1-step ahead ML model.

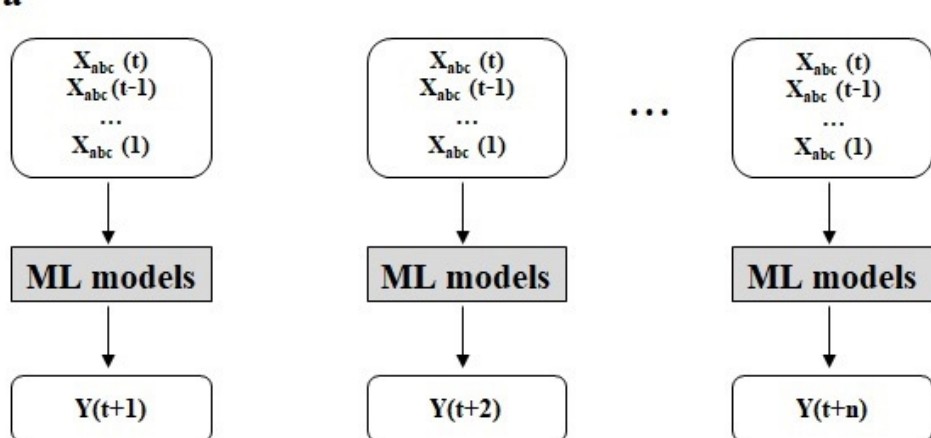

x : Input data, y : Output data (CH₄ yield), a: pH, b: Alkalinity, c : COD removal efficiency

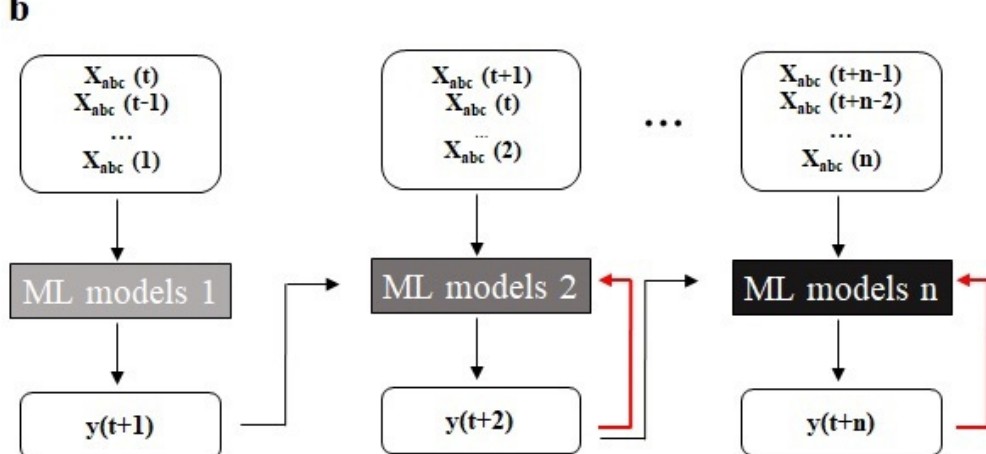

x : Input data, y : Output data (CH₄ yield), a: pH, b: Alkalinity, c : COD removal efficiency

**Figure 2.** Fundamentals of the (**a**) multi-step ahead and (**b**) 1-step ahead with retraining methods [21].

## 3. Results

### 3.1. Statistical Analysis

Figure 3a shows the results of PCA analysis when methane yield, pH, alkalinity, and COD removal efficiency of the BEAD reactor were applied as variables. The methane yield of the BEAD reactors shows positive correlations with the pH, alkalinity, and COD removal efficiency. The decreases in pH, alkalinity, and COD removal efficiency affected the decrease in the final methane yield [30]. In particular, pH had the highest correlation with the methane yield (BEAD reactor: 0.80), suggesting that rapidly overcoming the inhibition caused by a pH decrease could contribute to stable methane production (Figure 3b) [31]. The methane yield of the BEAD reactor showed no correlation with the VFAs, which did not satisfy the baseline value of KMO.

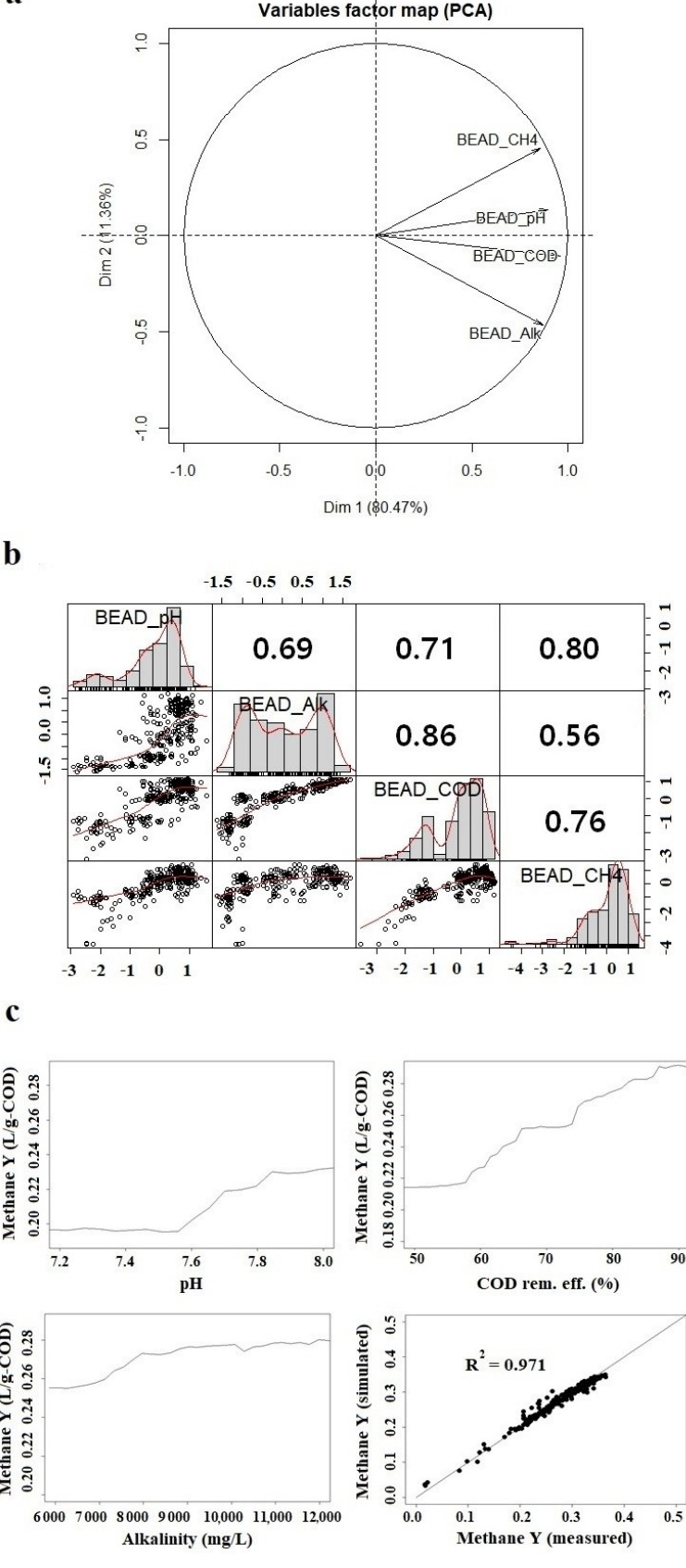

**Figure 3.** (**a**) Principal component analysis (PCA) loading plots and (**b**) scatter plots matrix for pairwise correlations, and (**c**) partial dependents correlations between independent variables (pH, alkalinity, and COD removal efficiency) and subordination variable (methane yield) of bio-electrochemical anaerobic digestion (BEAD).

When the three independent variables were used, the BEAD reactor showed the lowest RMSE values in the RFE-RE model. The variables were in the order of pH > COD removal efficiency > alkalinity for the BEAD reactor. The $R^2$ and RMSE of the BEAD reactor were 0.971 and 0.2382 L-CH$_4$/g-COD, respectively. This explained why the methane production of the BEAD reactor was affected by COD and H$^+$ consumption rates [22,32,33]. As shown in the partial correlations, the correlations were nonlinear and complex. The methane yield of BEAD decreased when the pH, alkalinity, and COD removal efficiencies were lower than 7.6, 8000 mg/L as CaCO$_3$, and 60%, respectively. The results of partial dependents correlations (Figure 3c) clearly showed that the methane yield of BEAD had non-linear relationships with pH, alkalinity, and COD removal efficiency, respectively, and clarified needs of enhanced prediction models for achieving high process stability in the BEAD operation.

### 3.2. Multi-Step Ahead ML Models

The RMSE value of the prediction result using the RNN method was 0.025 L-CH$_4$/g-COD, showing the best prediction efficiency (Figure 4). In addition, the RMSEs of RF, XGboost, LSTM, and SVR were 0.041, 0.053, 0.055, and 0.056 L-CH$_4$/g-COD, respectively. For the BEAD reactor, the prediction accuracy of the RNN method, which was effective for time-series prediction, was the highest. Therefore, RNN could reflect the characteristics of daily data appropriately, thereby showing a high prediction accuracy [34]. In cases of the BEAD reactor using the decision tree-based RF and XGboost, the prediction result was overestimated for the instantaneous methane yield decrease at the initial operation in each stage. This implied that the prediction accuracy was low for data that deviated significantly from the mean value of the regression calculated through learning [35]. Furthermore, the prediction efficiency of regression-based SVR, which assumed a linear combination of variables, was low in biological reactions with complex nonlinear relationships of various factors. For efficient operation and management of real BEAD reactors, it would be more effective to use the RNN method based on the accumulated time-series data when predicting the methane yield of the BEAD reactors with nonlinear relationships with time [36].

### 3.3. 1-Step Ahead ML Models

Reportedly, 1-step ahead prediction methods can predict and analyze time-series data with high accuracy and prediction efficiency [37,38]. Figure 5 shows the results of the 1-step ahead prediction using various ML models. In case of the BEAD reactor, the RMSE value of the prediction result using the RNN method was 0.017 L-CH$_4$/g-COD, showing the best prediction efficiency. The RMSEs of SVR, LSTM, RF, and XGboost were 0.021, 0.022, 0.028, and 0.030 L-CH$_4$/g-COD, respectively. In every ML models, The 1-step ahead with retraining method showed a higher RMSEs than the RMSEs of the multi-step ahead method shown earlier. This indicated that the 1-step ahead method which continuously retrains previous prediction values could more efficiently predict the methane yield of the BEAD reactor based on data that have nonlinear relationships with time [39]. In other words, because operation data are accumulated continuously in BEAD reactor that is operated continuously, the 1-step ahead method that facilitates learning by applying them in stages can be effectively applied [40]. In particular, the prediction accuracies of RF, XGboost, and SVR, which were not appropriate for time-series prediction, were increased through the 1-step ahead method, and they were not significantly different from the RMSE value of the RNN method. These results suggest that the prediction can be performed indirectly for the time-series data analysis using 1-step ahead method. Of note, the prediction value that deviates greatly from the regression section in the multi-step ahead prediction of decision tree-based RF and XGboost can be corrected based on the time-series learning and prediction through the 1-step ahead method. Therefore, the usability of the decision tree-based model can be increased in the prediction of nonlinear data over time [41,42].

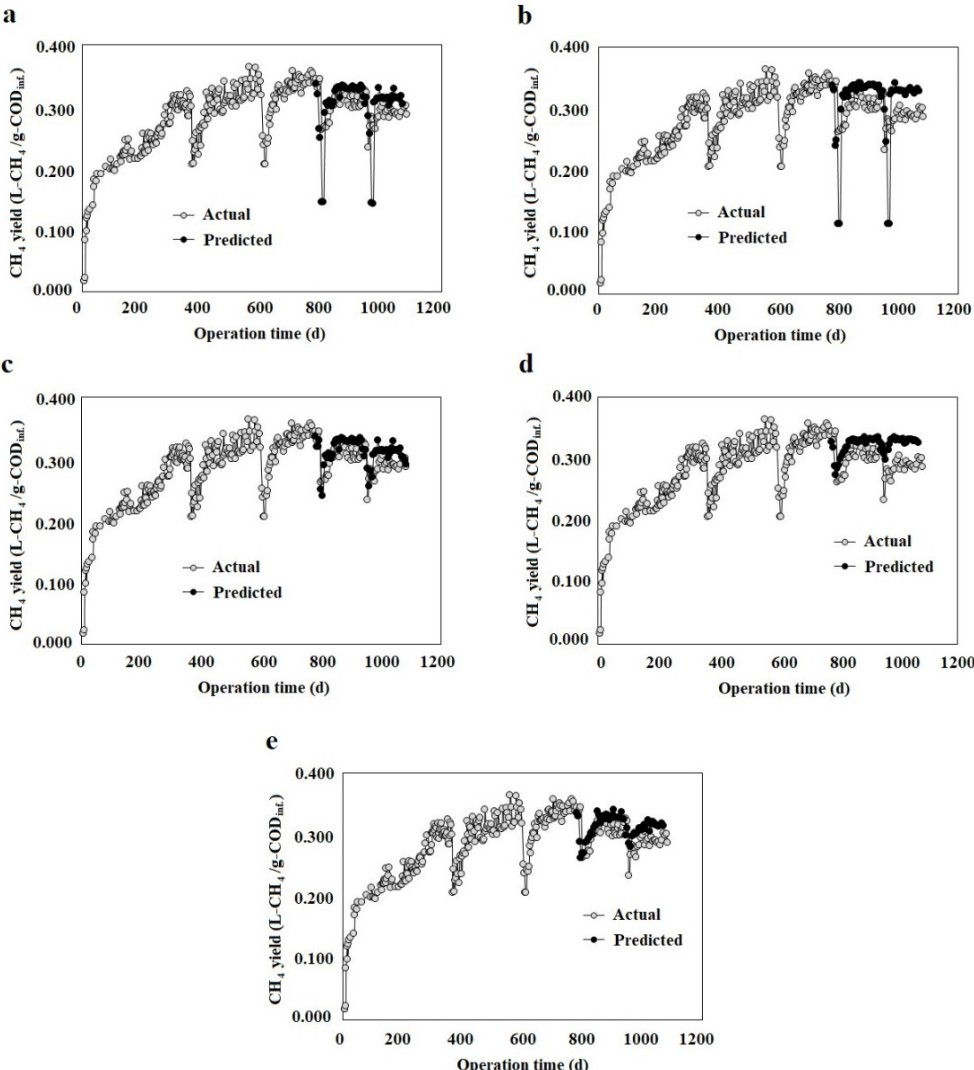

**Figure 4.** Results of the multi-step ahead predictions prediction of bio-electrochemical anaerobic digestion (BEAD) reactor using (**a**) random forest (RF), (**b**) extreme gradient boosting (XGboost), (**c**) support vector regression (SVR), (**d**) long short-term memory (LSTM), and (**e**) recurrent neural network (RNN) models.

### 3.4. Prediction of Methane Yield Using pH as Single Input Data

The 1-step ahead model using pH, alkalinity, and COD removal efficiency as input data was found to enable the effective prediction of time-series data. However, these input data are not available for real-time prediction because of the difficulty of prompt measurement in the full-scale BEAD process [43]. pH is the easiest parameter for monitoring full-scale BEAD processes using portable instruments and is one of the most important factors that directly affects methanogenic microorganism activity [44–46]. Therefore, the effect of pH as a single input data point on the prediction of methane yield was evaluated in this study. For the BEAD reactor, the prediction efficiency of the RNN method, which was effective for time-series prediction, was the highest. The 1-step ahead method of every ML model showed a higher prediction accuracy than the multi-step ahead prediction efficiency shown earlier. This indicated that the 1-step ahead method that facilitates learning by considering previous prediction values continuously could more efficiently predict the methane yield in the full-scale BEAD process based on the pH as a single input data [39]. Figure 6 shows the RMSE values of BEAD resulting from the multi-step and 1-step ahead RNN models that achieved the highest prediction efficiency. For the multi-step-ahead RNN model, the RMSE value of the BEAD reactor was 0.032 L-CH$_4$/g-COD. For the 1-step ahead RNN model, the

RMSE value of the BEAD reactor was 0.017 L-CH$_4$/g-COD. These results show that the methane yield could be effectively predicted by pH as a single input data and suggest the possibility of applying BEAD to a full-scale process [46].

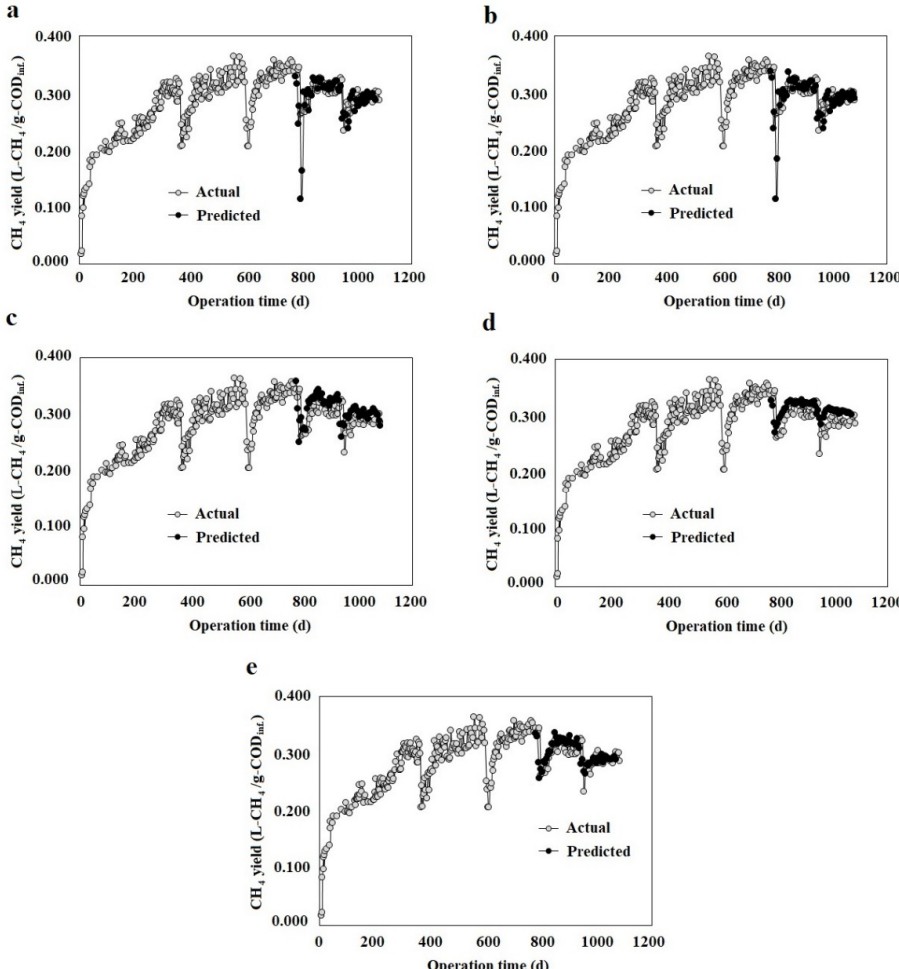

**Figure 5.** Results of 1-step ahead prediction of bio-electrochemical anaerobic digestion (BEAD) reactor using (**a**) random forest (RF), (**b**) extreme gradient boosting (XGboost), (**c**) support vector regression (SVR), (**d**) long short-term memory (LSTM), and (**e**) recurrent neural network (RNN).

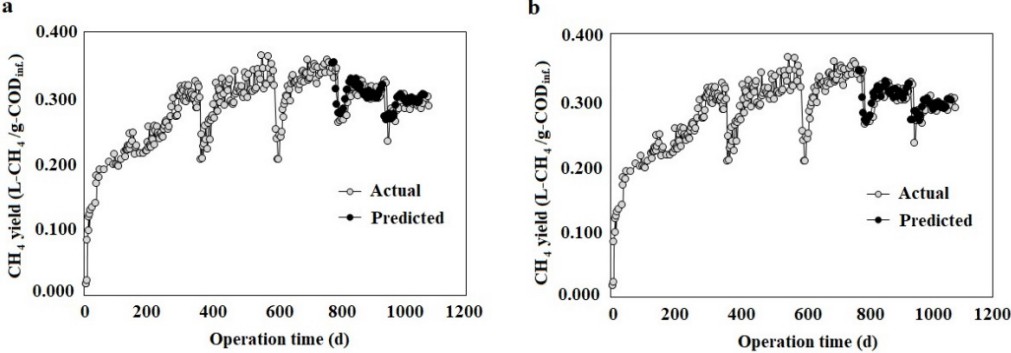

**Figure 6.** Results of multi-step ahead (**a**) and 1-step ahead (**b**) prediction of bio-electrochemical anaerobic digestion (BEAD) reactor using recurrent neural network (RNN) model with pH as a single input data.

## 4. Discussion

Results from the PCA showed that pH had the highest correlation with the methane yield in the BEAD reactor, which meant that quickly overcoming the inhibition caused

by a pH decrease could contribute to stable methane production. 1-step ahead prediction method could predict and analyze time-series data with high accuracy and prediction efficiency (Table 3). In other words, because operational performance data are continuously accumulated in the BEAD reactor, the 1-step ahead with retraining method that facilitates learning by applying them in stages can be effectively applied. The capability of the 1-step ahead with the retraining method could realize real-time monitoring and prediction of BEAD performance simultaneously. These potentials would be useful to achieve the stable operation of the full-scale BEAD process, especially when BEAD is faced with unexpected status, causing a loss of economic and energy production.

**Table 3.** RMSE values of BEAD reactor for multi-step ahead and 1-step ahead predictions using various machine learning models.

| Parameters | | RMSE (L-CH$_4$/g-COD) | | | | |
| --- | --- | --- | --- | --- | --- | --- |
| | | RF | XGboost | SVR | LSTM | RNN |
| BEAD | Multi-step ahead | 0.041 | 0.053 | 0.056 | 0.055 | 0.025 |
| | 1-step ahead | 0.028 | 0.030 | 0.021 | 0.022 | 0.017 |

RMSE: root mean square error, RF: random forest, COD: chemical oxygen demand, XGboost: extreme gradient boosting, SVR: support vector regression, LSTM: long short-term memory, RNN: recurrent neural network, BEAD: bio-electrochemical anaerobic digestion.

While alkalinity, COD removal efficiency, VFAs, and others could be also used as input parameters for ML models, they are not suitable for real-time predictions in full-scale BEAD processes due to time-consuming disadvantages and uneconomic applicability [21,43]. However, pH can be quickly analyzed by sensor-based portable detectors. Furthermore, pH is the most sensitive factor that directly affects methanogenic microorganism activity and methane yield [44–47], and change of pH showed the highest correlationship with BEAD performance in the statistic analysis of this study (Figure 3). Thus, the result of prediction using pH as a single input data showed that the methane yield could be effectively predicted by pH data and implied the possibility of practical application of BEAD, which could maintain optimum pH values via bio-electrochemical reactions (Table 4).

**Table 4.** RMSE values of BEAD reactor for multi-step ahead and 1-step ahead predictions using various machine learning models with pH as a single input data.

| Parameters | | RMSE (L-CH$_4$/g-COD) | | | | |
| --- | --- | --- | --- | --- | --- | --- |
| | | RF | XGboost | SVR | LSTM | RNN |
| BEAD | Multi-step ahead | 0.020 | 0.023 | 0.022 | 0.021 | 0.019 |
| | 1-step ahead | 0.019 | 0.022 | 0.019 | 0.019 | 0.017 |

RMSE: root mean square error, RF: random forest, COD: chemical oxygen demand, XGboost: extreme gradient boosting, SVR: support vector regression, LSTM: long short-term memory, RNN: recurrent neural network, BEAD: bio-electrochemical anaerobic digestion.

This study could show that the various ML models would be able to help BEAD achieves a higher process stability than AD. Moreover, 1-step ahead with the retraining methods could provide realizable applicability of various ML models to real world bio-processes. The pH could be realizable parameter as a single input data and its applicability was proven in this study. This possibility implies more detailed and scientific algorithm should be developed and modeled in the future.

**5. Conclusions**

This study confirmed that the 1-step ahead with the retraining method applied to various ML models was able to improve prediction accuracy of BEAD performance by retraining the prior state performances in the time series data. Notably, 1-step ahead with the retraining method significantly improved prediction accuracies during the OLR transition periods in the tree-based RF and regression-based SVR models. Another important finding of 1-step ahead method was that pH as only input parameter could be efficiently

used for real-time prediction of BEAD performance. The ML models using pH as a single input parameter were less accurate than those using multiple input parameters. However, pH was more efficient for monitoring than the other parameters, offering advantages in achieving real-time performance predictions for time-series full-scale operations.

**Supplementary Materials:** The following supporting information can be downloaded at: https://www.mdpi.com/article/10.3390/pr10010158/s1, Table S1: Input and output data for various ML models.

**Author Contributions:** Original draft preparation: A.C.; Conceptualization, data curation, and methodology: H.J. (Hangbae Jun) and J.S.; formal analysis and visualization: M.K. and H.J. (Heewon Jang); Conceptualization, review, and editing: J.P. All authors have read and agreed to the published version of the manuscript.

**Funding:** This research was supported by the Basic Science Research Program through the National Research Foundation of Korea (NRF) funded by the Ministry of Education (NRF-2021R1I1A3044486) and was supported by the Korea Ministry of Environment as Waste to Energy-Recycling Human Resource Development Project [YL-WE-19-001].

**Institutional Review Board Statement:** Not applicable.

**Informed Consent Statement:** Not applicable.

**Data Availability Statement:** Not applicable.

**Conflicts of Interest:** The authors declare no conflict of interest.

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
