# Peer review of "Application of Various Machine Learning Models for Process Stability of Bio-Electrochemical Anaerobic Digestion"

_processes, doi:10.3390/pr10010158_

Round 1

Reviewer 1 Report

This paper provides the results of using 1-step ahead machine learning to characterize the operation of a long-term lab scale bio-electrochemical anaerobic digestion operating on variable food waste substrate. Using five R program packages, the authors evaluated the collected dataset and then used 80% of the data as a training model for 1-step ahead machine learning models. The remaining 20% of the data was used for model validation.

No major errors or omissions were found. The major concern is that the dataset and working models are not included with the draft publication. This leaves open the issue of replication and potential follow up research by 3rd parties.

It is recommended that Figure 2, Figure 3, Figure 4, Figure 5, and Figure 6 all be revised to increase both the overall size of the sub-figures and increase the font size of axes, legends, and associated overlay text.

Author Response

Thank you for your valuable comments. The authors agreed with all of your suggestion and have made revisions referring your comments. Please see our responses and revision proofs in attached file. We described your comments by green, our responses by blue, and revision proofs by red. Hope our revisions would be acceptable.

Reviewer 2 Report

What is the novelty and originality of this work? Which should be clarified in the introduction

None reference from the Processes journal was added, therefore it does not present relevance with this journal

Emerging trends and future prospects section should be added to the document

More discussion about that first, 1-step ahead with the retraining method should be added and compared with other cases in the literature

Therefore, I cannot recommend the submitted manuscript is published in Processes in this way.

Author Response

(The authors gave the same response as above.)
